# Real-time profilometry by bicolor grating video projection

**Yujiao Zhang, Yiping Cao** * , **Haitao Wu** , **Haihua An, Xiuzhang Huang**

College of Electronics and Information Engineering, Sichuan University, Chengdu, China

* ypcao@scu.edu.cn

## Abstract

A novel real-time 2+1 three-dimensional(3D) measuring method based on bicolor grating video projection is proposed. Firstly, only two frames of bicolor gratings, in which the red channels are two sinusoidal fringes with a shifting phase of π/2 and the blue channels are the same background light equivalent to the DC component of the two sinusoidal fringes are encoded and arranged alternatively to synthesize into a repetitive bicolor grating video, While this video is projected onto the measured object, the real-time bicolor deformed pattern video can be recorded by using a color CMOS camera, and the bicolor deformed pattern sequence at different moments can be extracted by computer processing, so that the 2 +1 algorithm can be used to accomplish real-time 3D measurement of moving object. Before measuring, we used the same method to design two sinusoidal fringes with a difference of π in their red channels, respectively, to calibrate the sensitivity ratio between the red and blue channels of the CMOS camera, which can effectively eliminate the chromaticity imbalance between R and B channels and reduce the color crosstalk. Experimental results and analysis confirm the feasibility and effectiveness of the proposed method. Because the proposed method needs a repetitive bicolor grating video synthesized with only two-frame bicolor gratings to be projected, the 3D measurement acquisition speed and real-time accuracy will be improved compared with the traditional 2+1 3D measuring method.

## 1 Introduction

With the rapid development of science and technology in photoelectric information technology, image data processing technology, computer technology and other relevant fields, optical three-dimensional(3D) surface measurement technology has also been developed rapidly [1–3]. Nowadays, optical 3D surface measurement technology is applied in many fields such as mechanical engineering, industrial monitoring, computer vision, biomedicine, physical modeling, and so on [4–6]. The commonly used 3D surface measuring methods based on structured light illumination [7, 8] include Fourier transform profilometry (FTP) [9, 10] and phase measurement profilometry (PMP) [11, 12]. FTP requires only one frame of phase shift diagram, it is capable of retaining the fringe frequency in the frequency domain, removing high-frequency noise and carrier waves, recovering the fringe pattern from the frequency domain to the spatial domain via the inverse Fourier transform, reproducing the fringe field

**Data Availability Statement:** All relevant data are within the manuscript and its Supporting Information files. The minimal data set has been uploaded to GitHub: https://github.com/ananzhangyujiao/Experimental-data/releases.

**Funding:** This research is supported by the Special Grand National Project of China (under grant No. 2009ZX02204-008)

distribution, and lastly calculating the phase value of the fringe field [13, 14]. PMP usually uses three or more frames of phase shift diagrams to retrieve the phase information of the measured object. As opposed to FTP, PMP exhibits a lower speed but a higher measuring accuracy, a wider range of heights of measuring object, less effect by different reflectivity values on object surface, and less effect exerted by drastic variations or fractures on object surface [15, 16]. The modern 3D measurement is committed to investigating a higher measuring speed, higher measuring accuracy, as well as a wider measurement application range; thus, PMP is recognized as a more effective choice [17–21].

When testing objects, vibrations are likely to break the method because of the fuzziness and random phase errors caused by sequential exposures. To reduce the measurement error caused by its influence, Peter L. Wizinowich proposed a 2+1 phase shift algorithm [22, 23]. This method requires two phase-shifted $\pi/2$ fringe patterns and two phase-shifted $\pi$ fringe patterns, respectively. The first two are taken as the first and second frames of the common three-step PMP and the last two take the average as the third frame image. The proposed algorithm has the limited technical conditions and a narrow application scope [24]. The high-speed movement of objects will bring errors similar to vibration, to solve the error problem of moving objects, ZHANG S proposed an improved 2+1 phase shift algorithm [25]. The third frame directly projects the uniform plane illumination generated by the computer to replace the average effect of the third frame achieved by the two frames in the method Peter L.Wizinowich proposed. The traditional three-step PMP contains the phase information of the object in three pictures, respectively, however, in ZHANG S' improved 2+1 phase shift algorithm, the phase information of the object is only contained in two pictures. Since the sensitivity of the uniform plane graph to motion blur is significantly lower than that of the fringe graph, this method, compared with the conventional 2+1 phase shift algorithm, employs one frame less and exhibits a higher acquisition speed. At the same time, the digital projector projection directly eliminates the influence of nonlinear error of phase shifter, so ZHANG S' improved 2+1 phase shift algorithm can effectively reduce the measurement error caused by object motion.

In this paper, a novel bicolor 2+1 3D real-time measuring method based on grating video projection was proposed. First, only two frames of bicolor patterns are used, as against the three frames which are essential in the traditional 2+1 3D real-time measuring method. These two frames are encoded and arranged alternately to synthesize a repetitive bicolor grating video. Then the video is projected onto the object and the bicolor deformed patterns captured by the color camera that covered two frames of phase-shifting sinusoidal deformed patterns plus one background pattern could be extracted through the computer to reconstruct the 3D shape of the object measured. Thus, the 3D measurement acquisition speed and real-time measuring accuracy will be improved compared with the traditional 2+1 3D measuring method.

## 2 Principle

### 2.1 ZHANG S' 2 +1 phase shift algorithm

When a sinusoidal grating pattern is projected onto the surface of a 3D diffuse reflective object, the deformed pattern acquired from the imaging system may be represented as:

$$I_i(x, y) = R(x, y)\text{a} + R(x, y)\text{b}\cos[2\pi f_0 x + \varphi(x, y) + \delta_i] \tag{1}$$

In the Eq (1), $(x,y)$ represents the pixel point coordinates of the captured image, a expresses the component of DC term of the fringe, b expresses the fringe contrast; $R(x,y)$ expresses the reflectivity of different positions on the surface of the object; $\varphi(x,y)$ expresses the phase

modulated by the height of the measured object; $\delta_i$ represents the shifted phase of the i-frame fringe pattern.

ZHANG S' 2 +1 phase shift algorithm projects two frames of sinusoidal grating with phase shift $\pi/2$ and one frame of uniform background light with gray scale equal to the DC term of the sinusoidal grating. The two frames of deformed patterns and one frame of background image modulated by the surface shape of the object obtained from the imaging system can be expressed as:

$$I_1(x,y) = R(x,y)\text{a} + R(x,y)\text{b}\cos[2\pi f_0 x + \varphi(x,y)] \tag{2}$$

$$I_2(x,y) = R(x,y)\text{a} + R(x,y)\text{b}\sin[2\pi f_0 x + \varphi(x,y)] \tag{3}$$

$$I_3(x,y) = R(\text{x},y)\text{a}' \tag{4}$$

This method considers that the light intensity distribution of the background map collected by the camera is equal with the light intensity distribution of the DC term of the deformed fringe pattern, i.e., a = a'. From this, the difference between Eqs (2)–(4) respectively can obtain two light intensity distributions with only AC term. Finally, the expression of $\varphi(\text{x},y)$ can be obtained through calculation:

$$2\pi f_0 x + \varphi(x,y) = \arctan\frac{I_2(x,y) - I_3(x,y)}{I_1(x,y) - I_3(x,y)} \tag{5}$$

## 2.2 Theory of the proposed method

As we all know, the responsiveness of color CMOS camera to red, green, and blue light is different, and because the wavelength difference between red and blue light is relatively large, only red and blue color channels are used in this paper to reduce the phenomenon of color crosstalk. In practical application, the projected chromaticity and projected light intensity of the red and blue channels of the color projector [26] are different, and the sensitivity of the red and blue channels to light is also different when a color CMOS camera is collected. If two frames of red and blue uniform background lights with the identical value as the DC component of the sinusoidal fringe are separately projected, this can address the chromaticity imbalance problem attributed to the different sensitivities of the CMOS camera to red and blue. Subsequently, the red color of the identical color image in the practical experiment is ignored. To possibly eliminate the chromaticity imbalance attributed the different sensitivities of the CMOS camera to the red and blue lights, two bicolor gratings are designed in advance:

$$I_{\text{bi1}}(x,y) = [\text{a} + \text{b}\cos(2\pi f_0 x)]\overrightarrow{\boldsymbol{R}} + 0 \cdot \overrightarrow{\boldsymbol{G}} + \text{a}\overrightarrow{\boldsymbol{B}} \tag{6}$$

$$I_{\text{bi2}}(x,y) = [\text{a} - \text{b}\cos(2\pi f_0 x)]\overrightarrow{\boldsymbol{R}} + 0 \cdot \overrightarrow{\boldsymbol{G}} + \text{a}\overrightarrow{\boldsymbol{B}} \tag{7}$$

The red channel ($\overrightarrow{\boldsymbol{R}}$) in Eq (6) expresses the sinusoidal fringe, the blue channel ($\overrightarrow{\boldsymbol{B}}$) expresses the background light corresponding to the sinusoidal fringe; The red channel ($\overrightarrow{\boldsymbol{R}}$) in Eq (7) expresses a fringe pattern complementary to the sinusoidal fringe of Eq (6),the blue channel ($\overrightarrow{\boldsymbol{B}}$) expresses still the corresponding background light; The green channels ($\overrightarrow{\boldsymbol{G}}$) expresses all zero set to avoid interference to the red and blue channels as well cut the interference between the red and blue channels to some extent.

These two bicolor gratings are arranged alternatively to synthesize into a repetitive bicolor grating video. While this video is projected onto the reference plane, the fringe patterns of the

reference plane captured by the color CMOS camera can be expressed as:

$$I_{c1}(x, y) = R_r(x, y)[a + b\cos(2\pi f_0 x) + \varphi_0(x, y)]\overrightarrow{\boldsymbol{R}} + R_b(x, y)a\overrightarrow{\boldsymbol{B}} \qquad (8)$$

$$I_{c2}(x, y) = R_r(x, y)[a - b\cos(2\pi f_0 x) + \varphi_0(x, y)]\overrightarrow{\boldsymbol{R}} + R_b(x, y)a\overrightarrow{\boldsymbol{B}} \qquad (9)$$

$R_r$(x,y) and $R_b$(x,y) denote the reflectivity of the red channel and the blue channel, respectively, and $\varphi_0$(x,y) expresses the phase modulated by the reference plane. RGB tricolor extraction is carried out on the fringe image to obtain 2 monochromatic fringe patterns and 2 monochromatic background images:

$$I_{r1}(x, y) = R_r(x, y)[a + b\cos(2\pi f_0 x) + \varphi_0(x, y)] \qquad (10)$$

$$I_{r2}(x, y) = R_r(x, y)[a - b\cos(2\pi f_0 x) + \varphi_0(x, y)] \qquad (11)$$

$$I_{b1}(x, y) = R_b(x, y)a \qquad (12)$$

$$I_{b2}(x, y) = R_b(x, y)a \qquad (13)$$

$$(10) + (11): \qquad \frac{I_{r1}(x, y) + I_{r2}(x, y)}{2} = R_r(x, y)a \qquad (14)$$

$$(12) + (13): \qquad \frac{I_{b1}(x, y) + I_{b2}(x, y)}{2} = R_b(x, y)a \qquad (15)$$

As indicated from the in-depth analysis, $R_r$(x,y) and $R_b$(x,y) exhibit the similar characteristics. The ratio of reflectivity of the red channel and the blue channel is a constant. This is attributed to the chromaticity imbalance attributed to the different sensitivities of the CMOS camera to the red and blue lights. We introduce a chromaticity imbalance coefficient $K(x,y)$ caused by different sensitivities of CMOS camera to red and blue light, and we can obtain:

$$K(x, y) = \frac{I_{r1}(x, y) + I_{r2}(x, y)}{I_{b1}(x, y) + I_{b2}(x, y)} = \frac{R_r(x, y)}{R_b(x, y)} \qquad (16)$$

During real-time measurement, two other bicolor gratings with $\pi/2$ phase difference are arranged alternatively to synthesize into a new repetitive bicolor grating video. While this new video is projected onto measured object, the bicolor deformed patterns obtained from the imaging system can be expressed as follows:

$$I'_{c1}(x, y) = R'_r(x, y)[a + b\cos(2\pi f_0 x) + \varphi(x, y)]\overrightarrow{\boldsymbol{R}} + R'_b(x, y)a\overrightarrow{\boldsymbol{B}} \qquad (17)$$

$$I'_{c2}(x, y) = R'_r(x, y)[a + b\sin(2\pi f_0 x) + \varphi(x, y)]\overrightarrow{\boldsymbol{R}} + R'_b(x, y)a\overrightarrow{\boldsymbol{B}} \qquad (18)$$

Similarly, 2 monochromatic deformed patterns and 2 monochromatic background images can be extracted:

$$I'_{r1}(x, y) = R'_r(x, y)[a + b\cos(2\pi f_0 x) + \varphi(x, y)] \tag{19}$$

$$I'_{r2}(x, y) = R'_r(x, y)[a + b\sin(2\pi f_0 x) + \varphi(x, y)] \tag{20}$$

$$I'_{b1}(x, y) = R'_b(x, y)a \tag{21}$$

$$I'_{b2}(x, y) = R'_b(x, y)a \tag{22}$$

Considering the chromaticity imbalance coefficient $K(x,y)$ caused by different sensitivities of CMOS camera to red and blue light, it can be solved simultaneously as follows:

$$2\pi f_0 x + \phi(x, y) = \arctan \frac{I'_{r1}(x, y) - K(x, y)I'_{b1}(x, y)}{I'_{r2}(x, y) - K(x, y)I'_{b2}(x, y)} \tag{23}$$

The phase calculated by the Eq (23) is wrapped within the principal value range of the inverse trigonometric function $(-\pi, \pi]$ [27, 28]. The wrapped phase can be restored to a continuous phase through the phase unwrapping algorithm [29]. Finally, the 3D surface shape of the object can be reconstructed through the phase height mapping calculation of each point on the surface of the object.

## 3 Experimental results and analysis

To verify the measurement effect of the proposed method on real objects, an experimental system shown in Fig 1 is built. The digital projector used was the PLED-W200 model produced by ViewSonic. The camera used was DFM 72AUCO2, a color industrial camera produced by The Imaging Source. The size of the images captured by the camera is 640pixel×480pixel.

### 3.1 Comparison experiment

Before measuring the 3D shape of the measured object, we make the two-frame bicolor gratings $I_{bi1}(x,y)$ and $I_{bi2}(x,y)$ into a repetitive video and project it onto the reference plane through PLED-W200. Since we only need to use the red and blue channels in this paper, we set the green channel empty. The CMOS camera synchronously records the bicolor fringe pattern video in real time, so the bicolor fringe pattern sequences can be extracted through computer processing by selecting the corresponding two adjacent frames of the bicolor fringe patterns $I_{c1}(x,y)$, $I_{c2}(x,y)$, and the chromaticity imbalance coefficient can be calibrated in the pre-experimental procedure.

The two-frame bicolor gratings $I_{bi1}(x,y)$ and $I_{bi2}(x,y)$ are made into a repetitive video and projected onto a stationary face model object through PLED-W200. The bicolor deformed pattern video is collected synchronously in real time by the CMOS camera, and the experimental data of two adjacent frames of deformed patterns extracted by the computer are shown in Fig 2.

Fig 2(A) is the face model tested in the experiment. Fig 2(B) are bicolor grating patterns with a phase difference of $\pi/2$. Fig 2(C) and 2(D) are the corresponding two frames of bicolor deformed patterns. Fig 2(E) and 2(F) are monochromatic deformed patterns extracted from R channel respectively, and Fig 2(G) and 2(H) are background images extracted from B channel, respectively.

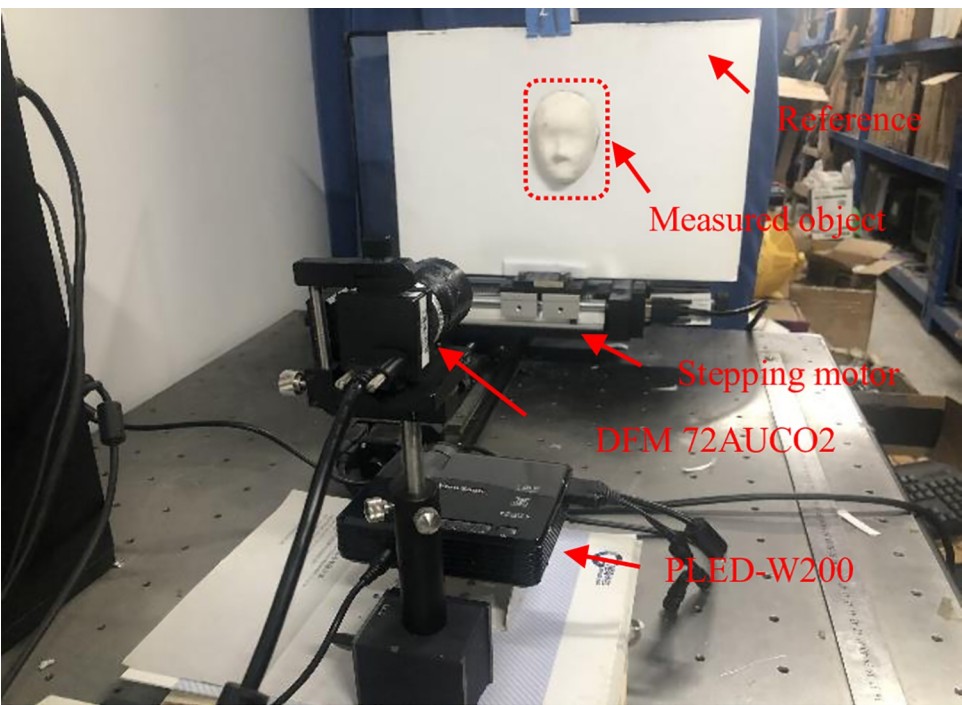

**Fig 1. Experimental system.**

As we all know, the result of traditional eight-step PMP has relatively high measuring accuracy [30], so the reconstruction result of traditional eight-step PMP is regarded as the quasi-truth in our experiments. Fig 3 is the experimental results. Fig 3(A) is a face model reconstructed by ZHANG S' 2+1 phase shift algorithm, and Fig 3(B) is a face model reconstructed by the bicolor 2+1 3D proposed method, both of which can form a better 3D face profile.

Fig 3(C)–3(E) are cross-sectional views of three methods including the traditional eight-step PMP, ZHANG S' 2+1 phase shift algorithm, and the proposed method. Dotted lines correspond to eight-step PMP, dash-dotted lines correspond to ZHANG S' 2+1 phase shift algorithm, and solid lines correspond to proposed method. Fig 3(C) is an overall cross-sectional view of the 319th column of the reconstructed face, Fig 3(D) is an enlarged view of the area A

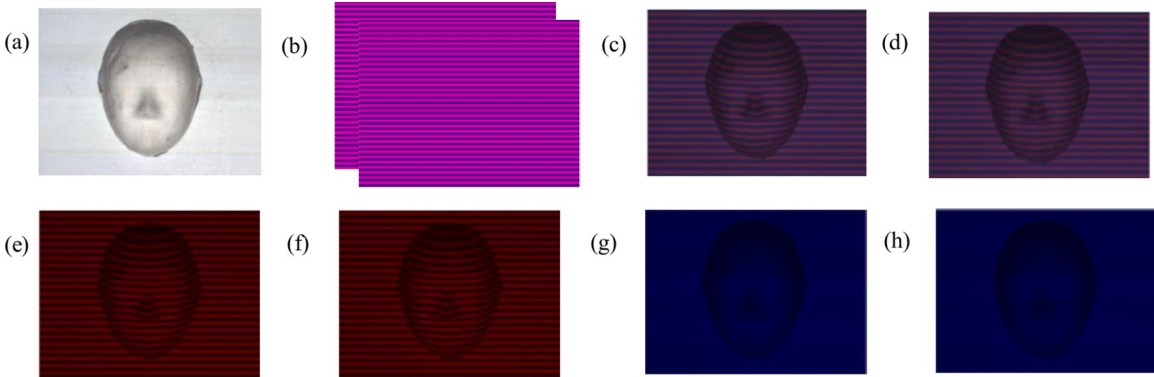

**Fig 2. Experimental data.** (a)Measured object, (b) grating pattern, (c)1st deform pattern, (d)2nd deform pattern, (e)1st R deform pattern, (f)2nd R deform pattern, (g) 1st B background image, (h) 2nd B background image.

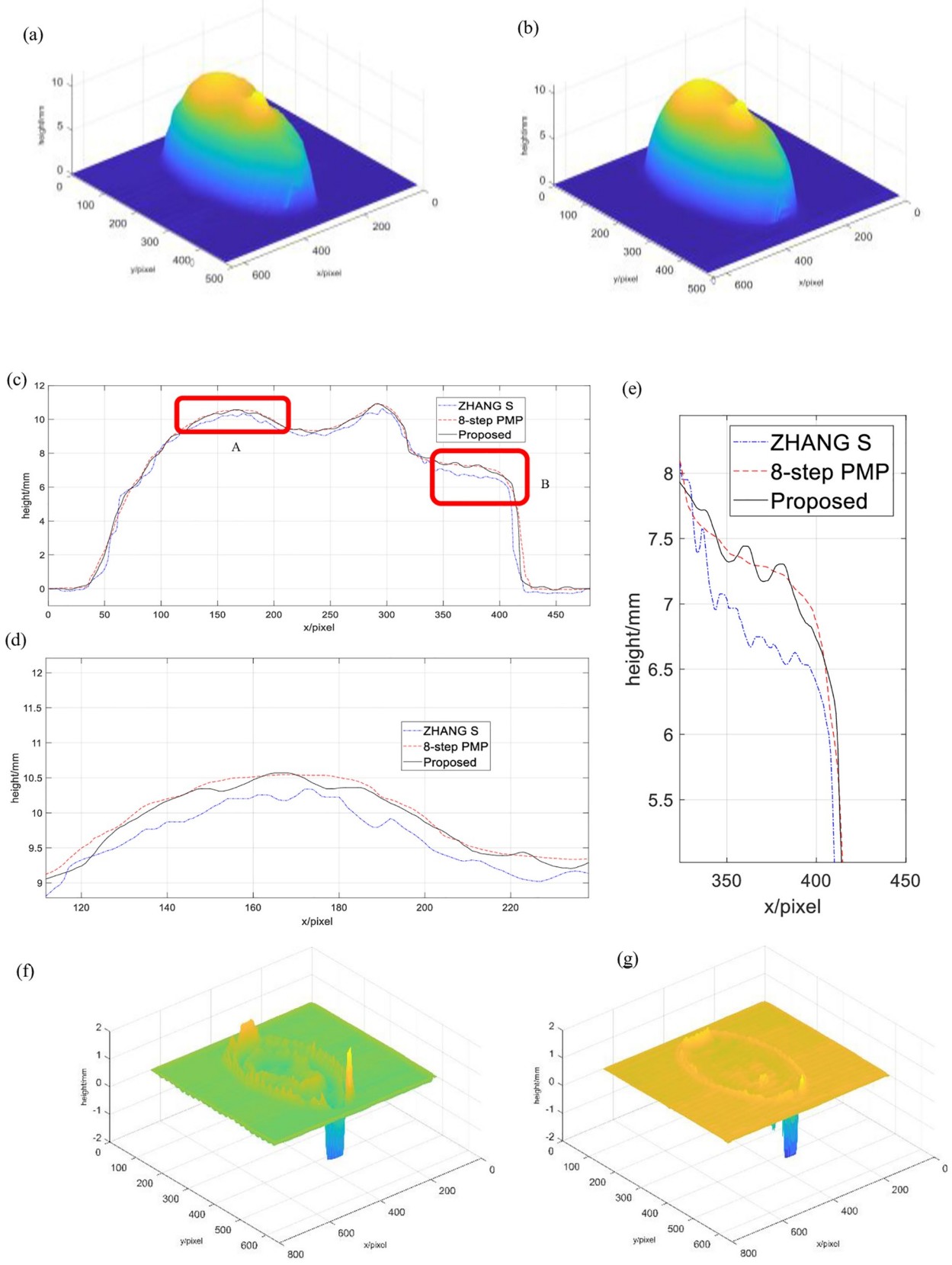

**Fig 3. Experimental results and height error distribution.** (a)Object reconstructed by ZHANG S′s algorithm, (b)Object reconstructed by proposed algorithm, (c)Sectional view of the 3D profile of the object, (d)Magnified view of the area A(e)Magnified view of area B, (f) Height error distribution of ZHANG S′s algorithm, (g) Height error distribution of proposed algorithm.

of Fig 3(C) and 3(E) is an enlarged view of area B of Fig 3(C). Analysis shows that the results of the two methods are close to the eight-step PMP, and the proposed method is closer to the traditional eight-step PMP. As opposed to the results obtained with the eight-step PMP results, the algorithm height error of ZHANG S as shown in Fig 3(F), and the algorithm height error of the proposed method is shown in Fig 3(G). It can be clearly seen that the height error of the proposed method is smaller. Calculating the height root mean square error of the two methods, the result of the algorithm of ZHANG S can be obtained as 0.1913, the proposed method can be obtained as 0.0629.

## 3.2 Real-time experiment

The experiment in this paper only needs to project two frames of bicolor gratings, which can effectively increase the speed of 3D measurement and improve efficiency. When the object moves online with the linear motion guide, if the projected two frames of bicolor grating are made into a periodic repetitive video, the PLED-W200 digital projector is used to project the video onto the surface of the online moving object. The bicolor deformed pattern video is synchronously recorded in real time by the color CMOS camera, and the bicolor deformed pattern sequences at different times are extracted by computer processing, to accomplish real-time online 3D measurement of the moving object.

Here, a double-layer rectangular model moving online is measured in real-time. By starting the stepper motor to control the heart model moving online, DLP stably projects the bicolor grating video onto the double-layer rectangular model. The bicolor deformed patterns at different times are simultaneously collected by the CMOS camera, and each group of deformed patterns are captured at 30 frames per second (30fps), are shown in S1 Visualization. The part of the experimental results are shown in Fig 4.

Fig 4(A)–4(C) respectively show the bicolor deformed patterns extracted by the real-time online moving double-layer rectangular model under the conditions of state 1, 2, and 3 respectively. Fig 4(D)–4(F) respectively show the monochrome deformed patterns extracted by the real-time online moving double-layer rectangular model under the corresponding state conditions. Fig 4(G)–4(I) respectively show the 3D shapes reconstructed by the real-time online moving double-layer rectangular model under the corresponding time conditions, more reconstructed results are shown in S2 Visualization. The 3D topography of the real-time moving double-layer rectangular model in different time states can be well reconstructed. Thus, it proves that the proposed method is suitable for real-time 3D measurement.

In order to further verify the measuring accuracy of the proposed method, we here take the traditional eight-step PMP result of measuring the static double-layer rectangular model as the quasi-truth and conduct a real-time online measurement comparison experiment with the ZHANG S' method. Before the experiment, the measured object is accurately positioned and controlled at a specific position by a stepping motor. The traditional eight-step PMP is used to perform a static measurement at the specific position to obtain the quasi-truth of the reconstructed surface of the measured object, and then the steps are precisely controlled. When the electric motor moves the measured object to a specific position, real-time online measurements of the two methods are carried out to ensure the comparability of the reconstruction results of different methods. The comparison result is shown in Fig 5.

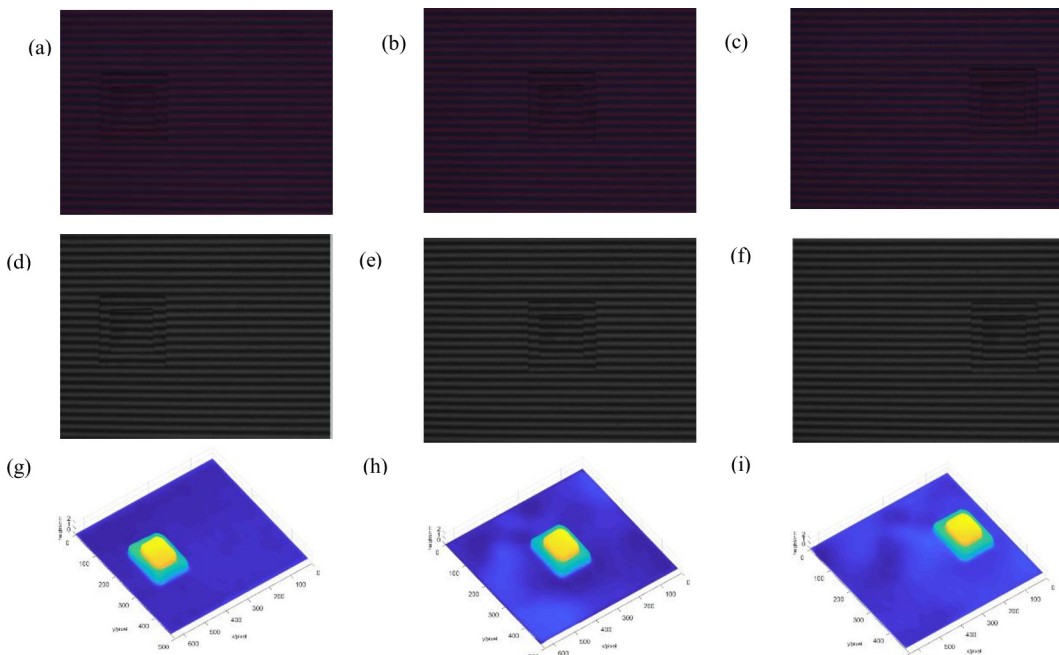

**Fig 4. Experimental results of real-time 3D measurement for moving double-layer rectangular model.** (a)-(c) 1st-3rd deform patterns, (d)-(f) 1st-3rd monochromatic deform pattens, (g)-(i)objects reconstructed of time 1 to 3.

Fig 5 is a cross-sectional view of a reconstructed double-layer rectangular model 3D surface, in which the dotted lines correspond to eight-step PMP, the dash-dotted lines correspond to ZHANG S' 2+1 phase shift algorithm, and the solid lines correspond to proposed method. Fig 5(A) is an overall cross-sectional view of the 250th row of the reconstructed double-layer rect-angular model 3D surface, Fig 5(B) is an enlarged view of the heart model A of Fig 5(A) and 5(C) is an enlarged view of the part B of Fig 5(A) and 5(D) is an enlarged view of the part C of Fig 5(A). Analysis shows that, the results of the proposed method are close to the eight-step PMP, ZHANG S' method is not effective in restoring the middle connection of the double-layer rectangle. From Fig 5(B) and 5(D), due to the time difference between real-time motion, real-time projection and capture of the object, the object is slightly deformed during imaging, and the real-time bicolor 2+1 image is closer to the static real result, which shows that the bicolor 2+1 projection of two frames has better real-time performance than the ZHANG S' phase shift algorithm of three-frame projection. As opposed to the results obtained with the eight-step PMP results, the real-time algorithm height error of ZHANG S as shown in Fig 5(E), and the real-time algorithm height error of the proposed method is shown in Fig 5(F). It can be clearly seen that the height error of the proposed method is smaller. Calculating the height root mean square error of the two methods, the result of the algorithm of ZHANG S can be obtained as 0.1055, the proposed method can be obtained as 0.0517.

According to the above comparison method, we select the speed as 24mm/s, 30mm/s, and 36mm/s respectively by controlling the speed of the stepper motor to complete real-time experiment. Then, we calculate the corresponding root mean square errors and made a table as seen in Table 1."

It can be observed that the measuring effect becomes worse as the speed increases. The rea-son for this result is that the higher the speed of the object was, the greater the amplitude of the object's vibration would be, the more factors that the captured image would interfere with, the

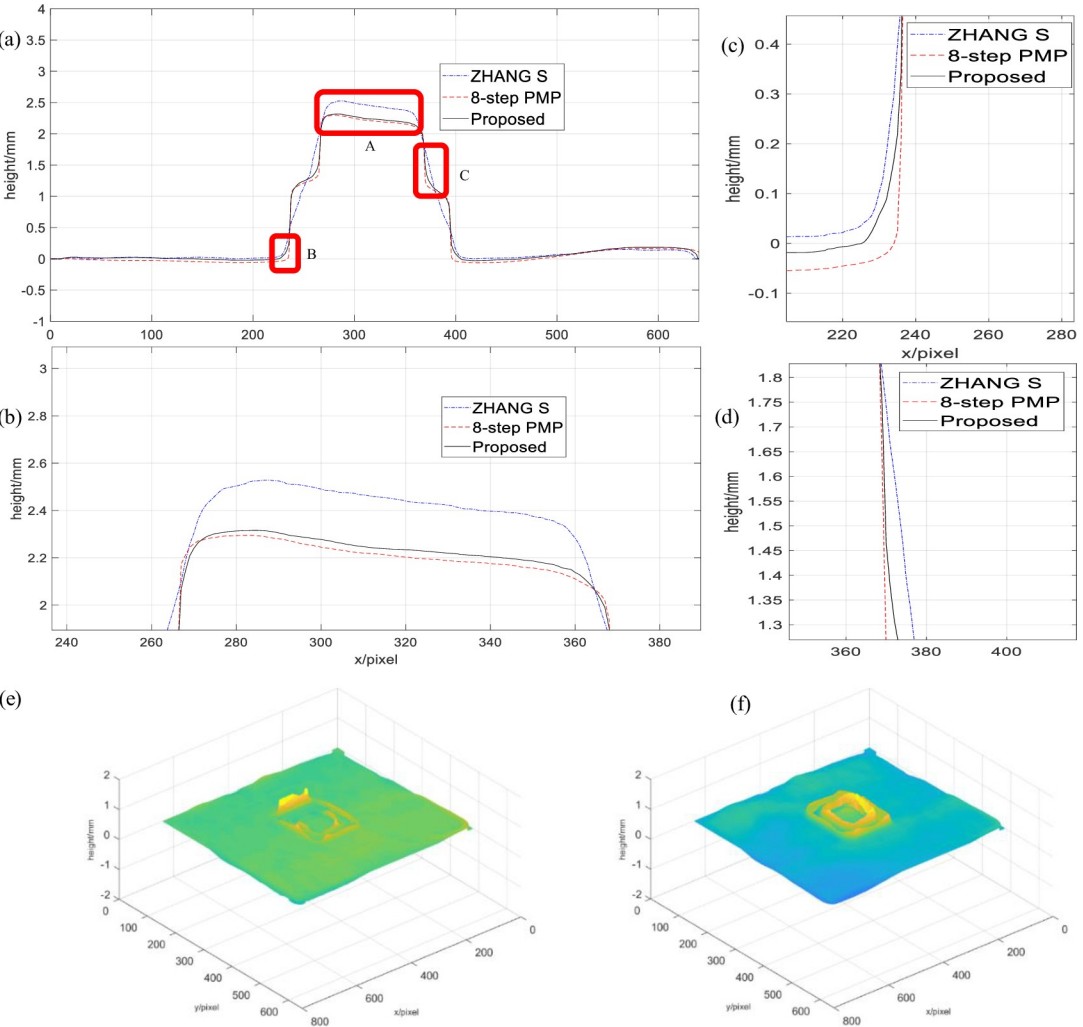

**Fig 5. The 235th row comparison results of the reconstructed object.** (a) Sectional view of the 3D profile of the object, (b) Magnified view of the A area, (c) Magnified view of the B area(d) Magnified view of the C area, (e) Height error distribution of ZHANG S′s algorithm, (f) Height error distribution of proposed algorithm.

more unstable resulting distorted image will be and the worse the measurement effect would be.

In order to further prove the feasibility of this method, a more complex sector model was measured in real time according to the same method. The bicolor deformed patterns at different times are simultaneously collected by the CMOS camera, and each group of deformed patterns are captured at 30 frames per second (30fps), are shown in S3 Visualization. The part of the experimental results is shown in Fig 6.

**Table 1. The height root mean square error at different speeds.**

| Speed(mm/s)<br>method | 24mm/s | 30mm/s | 36mm/s |
|---|---|---|---|
| ZHANG S | 0.0817 | 0.1055 | 0.1443 |
| Proposed method | 0.0410 | 0.0517 | 0.0706 |

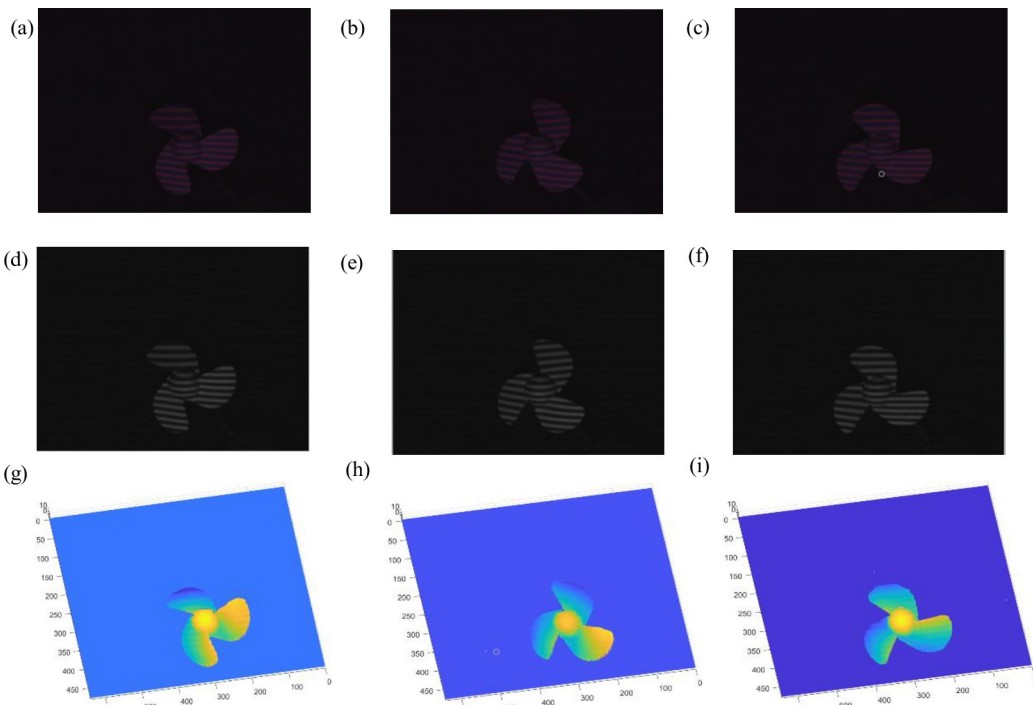

**Fig 6. Experimental results of real-time 3D measurement for moving sector model.** (a)-(c) $1^{st}$-$3^{rd}$ deform patterns, (d)-(f) $1^{st}$-$3^{rd}$ monochromatic deform pattens, (g)-(i)objects reconstructed of time 1 to 3.

Fig 6(A)–6(C) respectively show the bicolor deformed patterns extracted by the real-time online moving sector model under the conditions of state 1, 2, and 3 respectively. Fig 6(D)–6(F) respectively show the monochrome deformed patterns extracted by the real-time online moving sector model under the corresponding state conditions. Fig 6(G)–6(I) respectively show the 3D shapes reconstructed by the real-time online moving sector model under the corresponding time conditions, all reconstructed results are shown in S4 Visualization. It can be observed that the 3D reconstruction of the sector model has also been well restored, and its rotating tense can be restored in real time, which further proves the real-time feasibility of the proposed method.

## 4 Conclusion

In this paper, a novel 2+1 3D real-time measuring method based on bicolor grating video projection is proposed. Compared with the traditional 2+1 3D real-time measurement, which requires three frames of patterns, the proposed method only needs to encode two frames of bicolor grating and arranged alternatively to synthesize into a repetitive bicolor grating video, thus shortening the 3D measurement time and effectively improving the real-time performance of 2+1 measurement. By introducing the chromaticity imbalance coefficient between RGB channels, after the deformation fringe is calibrated, the error caused by chromaticity imbalance is effectively reduced on the one hand, and the color crosstalk problem is also alleviated on the other hand. The validity and feasibility of this method are verified by experiments, and it has a good application prospect in real-time 3D measurement. Of course, this method has certain limitations for measuring three-dimensional objects with colored surfaces.

## Supporting information

**S1 Visualization. Moving double-layer rectangular model experiment.** The 12 frames of deformed patterns collected during the movement of the double-layer rectangular model are extracted and made into a gif.
(GIF)

**S2 Visualization. Reconstruction results of moving double-layer rectangular model.** The 12 frames of the 3D shapes reconstructed by the real-time online moving double-layer rectangular model under the corresponding moving states and made into a gif.
(GIF)

**S3 Visualization. Moving sector model experiment.** The 10 frames of deformed patterns collected during the movement of the sector model are extracted and made into a gif.
(GIF)

**S4 Visualization. Reconstruction results of moving sector model.** The 10 frames of the 3D shapes reconstructed by the real-time online moving sector model under the corresponding moving states and made into a gif.
(GIF)

## Author Contributions

**Conceptualization:** Yujiao Zhang, Yiping Cao.

**Data curation:** Yujiao Zhang, Yiping Cao.

**Formal analysis:** Yujiao Zhang.

**Funding acquisition:** Yiping Cao.

**Investigation:** Haihua An, Xiuzhang Huang.

**Methodology:** Yiping Cao.

**Project administration:** Yiping Cao.

**Resources:** Yiping Cao.

**Software:** Yujiao Zhang.

**Supervision:** Yujiao Zhang, Yiping Cao, Xiuzhang Huang.

**Validation:** Yujiao Zhang, Yiping Cao, Haitao Wu.

**Visualization:** Yujiao Zhang.

**Writing – original draft:** Yujiao Zhang.

**Writing – review & editing:** Yujiao Zhang, Yiping Cao.

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
