## [Decision Letter · Decision Letter 0]

12 Apr 2021

PONE-D-21-03909

Real-time profilometry by bicolor grating video projection

PLOS ONE

Dear Dr. Cao,

Thank you for submitting your manuscript to PLOS ONE. After careful consideration, we feel that it has merit but does not fully meet PLOS ONE’s publication criteria as it currently stands. Therefore, we invite you to submit a revised version of the manuscript that addresses the points raised during the review process.

We look forward to receiving your revised manuscript.

Kind regards,

Ireneusz Grulkowski, PhD

Academic Editor

PLOS ONE

Journal Requirements:

[This research is supported by the Special Grand National Project of China (under grant No. 2009ZX02204-008).]

 [The funders had no role in study design, data collection and analysis, decision to publish, or preparation of the manuscript.]

Reviewers' comments:

Reviewer's Responses to Questions

**Comments to the Author**

1. Is the manuscript technically sound, and do the data support the conclusions?

Reviewer #1: Yes

Reviewer #2: No

2. Has the statistical analysis been performed appropriately and rigorously? 

Reviewer #1: Yes

Reviewer #2: No

3. Have the authors made all data underlying the findings in their manuscript fully available?

Reviewer #1: Yes

Reviewer #2: No

4. Is the manuscript presented in an intelligible fashion and written in standard English?

Reviewer #1: No

Reviewer #2: No

5. Review Comments to the Author

Reviewer #1: This paper proposed a novel real-time 2+1 three-dimensional measuring method based on a two-color fringe projection. In general, the structure of this paper is clear, and there are some new ideas especially on the fusion and phase recovery of two-color fringes. But the article still has many problems before published.

1. The English writing needs to be revised, because there are too many wording mistakes which influence the readability of the article. For example, in line 40, ‘faster speed’ should be placed by ‘higher speed’; in line 41, ‘wider measurement application range’, the meaning is not clear; in line 48, ‘application scope is very narrow’; many wrong uses of ‘Figs’ in line 170…; in line 174 wrong use of ‘quasi-true’; in line 222, wrong use of ‘cow’; invalid format in line 170 ‘π/2’, etc.

2. There are many descriptions hard to understand. For example, line 33-36; line 38-40; line 97-99; line 123-125…

3. Many references are miscoded. For example, line 31, 36, 42…

4. Suggest that the red number should be placed in the Fig.1.

5. About the experiment, there was no quantitative analysis in the comparative experiment. It will be better to give the residual figure between each of the method and measurement accuracy values.

6. I still have a question why the eight-step PMP was used for a reference not the five-step or the other method?

7. High speed measurements are expected to give speed values and comparisons.

At last, this paper should be revised on the English writing and format. I suggest the paper should be polished. The experiments should be richer. The accuracy values should be given as well as the speed values.

Hope the suggestions helps you.

Reviewer #2: The experimental section does not specify if the data is cherry-picked or averaged or repeatable over several frames. Unfortunately, the manuscript contains lots of English errors, and also, references are displayed as "Error! Reference source not found."

6. PLOS authors have the option to publish the peer review history of their article (what does this mean?). If published, this will include your full peer review and any attached files.

Reviewer #1: **Yes: **Fumin Zhang

Reviewer #2: No

---

## [Author Response · Author response to Decision Letter 0]

11 May 2021

Dear Editor and Reviewers: 

Thank you very much for your excellent editing work of our manuscript entitled “Real-time profilometry by bicolor grating video projection ([PONE-D-21-03909]-[EMID:a2a144e40538be21])”. And we also appreciate the reviewers for their valuable comments about our manuscript. Those comments are all valuable and very helpful for revising and improving our manuscript. We have studied the comments carefully and have made the correction which we hope to be able to meet the approval. The Response to Reviewer and Editor Comments is attached below. Hope our revised manuscript meets the high publication standard set by PLOS ONE.

Sincerely,

Yujiao Zhang, Yiping Cao, Haitao Wu, Haihua An, Xiuzhang Huang Sichuan University

Reviewer 1:

Comments to the Author

This paper proposed a novel real-time 2+1 three-dimensional measuring method based on a two-color fringe projection. In general, the structure of this paper is clear, and there are some new ideas especially on the fusion and phase recovery of two-color fringes. But the article still has many problems before published.

1. The English writing needs to be revised, because there are too many wording mistakes which influence the readability of the article. For example, in line 40, ‘faster speed’ should be placed by ‘higher speed’; in line 41, ‘wider measurement application range’, the meaning is not clear; in line 48, ‘application scope is very narrow’; many wrong uses of ‘Figs’ in line 170…; in line 174 wrong use of ‘quasi-true’; in line 222, wrong use of ‘cow’; invalid format in line 170 ‘π/2’, etc.

Response:

Thank you for raising this question, and I am sorry that my English grammatical errors and improper expressions have caused you difficulties in reading. I have corrected the grammar of the thesis from beginning to end. After receiving the modification comments, I did the real-time experiment again. And this time, I selected the 250th column of the reconstructed heart, so I changed the corresponding part of the experiment expression accordingly.

Revision:

1) In page 2 line 38, “slower speed” has been revised as “a lower speed”.

2) In page 2 line 38, “higher measuring accuracy, wider measurable range” has been revised as “a higher measuring accuracy, a wider range of heights of measuring object”.

3) In page 2 line 41, “faster measurement speed, higher measuring accuracy, and wider measurement application range” has been revised as “a higher measuring speed, higher measuring accuracy, a wider measuring application range”.

4) In page 9 line 172, “Figs 2(c) and 2(d)” has been revised as “Fig 2(c) and Fig 2(d)”.

5) In page 9 line 173, “Figs 2(e) and 2(f)” has been revised as “Fig 2(e) and Fig 2(f)”.

6) In page 9 line 174, “Figs 2(g) and 2(h)” has been revised as “Fig 2(g) and Fig 2(h)”.

7) In page 10 line 181, “Figs 3(c)-(e)” has been revised as “Fig 3(c)–Fig3 (e)”.

8) In page 12 line 208, “Figs 4 (a)-(c)” has been revised as “Fig 4(a)-Fig 4(c)”.

9) In page 12 line 209, “Figs 4(d)-(f)” has been revised as “Fig 4(d)-Fig 4 (f)”.

10) In page 12 line 211, “Figs 4(g)-(i)” has been revised as “Fig 4(g)-Fig 4(i)”.

11) In page 14 line 231, “Figs 5(b) and 5(d)” has been revised as “Fig 5(b) and Fig 5(d)”.

12) In page 9 line 176, page 12 line 216, page 13 line 220 “quasi-true value” has been revised as “quasi-truth”.

13) In page 13 between line 223 and 224, “323th cow” has been revised as “250th column”.

14) In page 14 line 228, “323th cow” has been revised as “250th column”.

15) In page 7 line 131, page 9 line 172 “π/2” has been revised as “ ”.

16) In page 3 line 61, “is” has been revised as “was”.

17) In page 3 line 61, “Firstly” has been revised as “First”.

18) In page 3 line 62, “are” has been revised as “were”.

19) In page 3 line 63, “is” has been revised as “was”.

20) In page 3 line 64, “containing” has been revised as “that covered”.

21) In page 3 line 65, “can” has been revised as “could”.

22) In page 3 line 66, “measured object” has been revised as “object measured”.

23) In page 4 line 73, page 4 line 74, page 4 line 75, page 5 line 104, page 5 line 105, page 5 line 106, page 5 line 107, page 6 line 115, “is” has been revised as “expresses”.

24) In page 6 line 114, “are” has been revised as “denote”.

25) In page 4 line 86, “expressions (2) and (3) and (4)” has been revised as “Eq (2), Eq (3) and Eq (4)”.

26) In page 6 line 114, we add two words “the”.

2. There are many descriptions hard to understand. For example, line 33-36; line 38-40; line 97-99; line 123-125…

Response: 

I am sorry that I did not express clearly enough in these places, which caused your reading difficulties. I re-expressed the content of these places.

Revision:

1) In page 2 line 32-36, The content “FTP requires only one frame of phase shift diagram, retains the fringe frequency in the frequency domain, removes high frequency noise and carrier waves, recovers the fringe pattern from the frequency domain to the spatial domain through inverse Fourier transform, reproduces the fringe field distribution, and finally calculates the phase value of the fringe field [13-14].” has been revised as “FTP requires only one frame of phase shift diagram, It is capable of retaining the fringe frequency in the frequency domain, removing high-frequency noise and carrier waves, recovering the fringe pattern from the frequency domain to the spatial domain via the inverse Fourier transform, reproducing the fringe field distribution, and lastly calculating the phase value of the fringe field [13-14]”.

2) In page 2 line 37-42, The content “Compared with FTP, PMP has slower speed but higher measuring accuracy, wider measurable range, less influence by different reflectivity on the surface of the object, and less influence by drastic changes or fractures on the surface of the object[15-16]. Modern 3D measurement is devoted to the research of faster measurement speed, higher measuring accuracy, and wider measurement application range, so PMP is a better choice[17-21] .” has been revised as “As opposed to FTP, PMP exhibits a lower speed but a higher measuring accuracy, a wider measurable range, less effect by different reflectivity values on object surface, and less effect exerted by drastic variations or fractures on object surface[15-16]. The modern 3D measurement is committed to investigating a higher measuring speed, higher measuring accuracy, as well as a wider measurement application range; thus, PMP is recognized as a more effective choice [17-21].”.

3) In page 3 line 48-49, I corrected the sentence “The technical conditions of the proposed method are limited and the application scope is very narrow[24]”to “The proposed algorithm has the limited technical conditions and a narrow application scope [24]”.

4) In page 3 line 55-57, this sentence is a bit difficult to read, so I corrected “Because the sensitivity of the uniform plane graph to motion blur is much lower than that of the fringe graph, moreover, compared with the traditional 2+1 phase shift algorithm, this method uses less one frame of image and has faster acquisition speed.” to “Since the sensitivity of the uniform plane graph to motion blur is significantly lower than that of the fringe graph, this method, compared with the conventional 2+1 phase shift algorithm, employs less one frame of image and exhibits a higher acquisition speed.”

5) In page 5 line 96-99, The content “If two frames of red and blue uniform background light with the same value as the DC component of the sinusoidal fringe are separately projected to solve the chromaticity imbalance problem caused by the different sensitivity of the CMOS camera to red and blue, then the red color of the same color image in the actual experiment is ignored..” has been revised as “If two frames of red and blue uniform background lights with the identical value as the DC component of the sinusoidal fringe are separately projected to address the chromaticity imbalance problem attributed to the different sensitivities of the CMOS camera to red and blue. Subsequently, the red color of the identical color image in the practical experiment is ignored. To possibly eliminate the chromaticity imbalance attributed the different sensitivities of the CMOS camera to the red and blue lights, two bicolor gratings are designed in advance:”.

6) In page 6 line 124-127, The content “Further analysis shows that and have similar characteristics. It can be seen that the ratio of reflectivity of red channel and blue channel is in constant characteristic. This is caused by the chromaticity imbalance caused by the different sensitivities of CMOS camera to red and blue light.” has been revised as “As indicated from the in-depth analysis, and exhibit the similar characteristics. which demonstrates that the ratio of reflectivity of the red channel and the blue channel exhibits the constant characteristic. This is attributed to the chromaticity imbalance attributed to the different sensitivities of the CMOS camera to the red and blue lights.”.

3. Many references are miscoded. For example, line 31, 36, 42…

Response:

Thank you so much for your question, and I am sorry that my carelessness caused this format problem. I have corrected the reference number hyperlink problem. 

Revision:

There are three numbering errors in the preview state. 

1) In page 2 line 31, Number “[7-8]” has updated the number hyperlink；

2) In page 2 line 36, Number “[13-14]” has updated the number hyperlink；

3) In page 2 line 42, Number “[17-21]” has updated the number hyperlink；

4. Suggest that the red number should be placed in the Fig.1.

Response:

Thank you very much for your valuable comments. After my consideration, I also think that your suggestions for modification are better. 

Revision:

So I replace the Fig.1 

between lines 155 and 156 in page 8 with the following figure

5. About the experiment, there was no quantitative analysis in the comparative experiment. It will be better to give the residual figure between each of the method and measurement accuracy values.

Response:

Thank you so much for your suggestion. I am sorry that I did not do the quantitative analysis of the experiment. Now I am supplementing the quantitative analysis of the experiment. Compared with the eight-step PMP, ZHANG S and the proposed method are compared to obtain the height error distribution and calculate the height root mean square error. We did a set of static experiments again We respectively added the height error distribution of the ZHANG S algorithm (Fig3 (f)) and the proposed method relative to the eight-step PMP (Fig3 (g)) to Fig.3 in the static face model measurement. Then, we separately calculated the height root mean square error relative to the eight-step PMP, the result of the algorithm of ZHANG S can be obtained as 0.1913, the proposed method can be obtained as 0.0629. 

In the same way, we also did a set of real-time experiments again, and the cross-sectional view is replaced with the 250th column. We respectively added the height error distribution of the ZHANG S algorithm (Fig.5(e)) and the proposed method relative to the eight-step PMP(Fig.5(f)) to Fig.5 in the real-time heart model measurement.

Revision:

In page 10, between line 177 and line 178, the measurement results of the face model at rest. 

The content of the analysis is added between line 187 and 192 in page 11, the content presents “As opposed to the results obtained with the eight-step PMP results, the algorithm height error of ZHANG S as shown in Fig 3(f), and the algorithm height error of the proposed method is shown in Fig 3(g). It can be clearly seen that the height error of the proposed method is smaller. Calculating the height root mean square error of the two methods, the result of the algorithm of ZHANG S can be obtained as 0.1913, the proposed method can be obtained as 0.0629.”.

In page 13, between line 223 and line 224, the measurement results of the heart model at real time. 

The content of the analysis is added between line 235 and 239 in page 14, the content presents “As opposed to the results obtained with the eight-step PMP results, the real-time algorithm height error of ZHANG S as shown in Fig 5(e), and the real-time algorithm height error of the proposed method is shown in Fig 5(f). It can be clearly seen that the height error of the proposed method is smaller. Calculating the height root mean square error of the two methods, the result of the algorithm of ZHANG S can be obtained as 0.1546, the proposed method can be obtained as 0.0513.”.

6. I still have a question why the eight-step PMP was used for a reference not the five-step or the other method?

Response:

Thank you so much for asking this question, and I am sorry for not being able to describe in detail the advantages of 8-step PMP as a quasi-truth method in my article. The reason for using this method is that I read a thesis called “Gamma model and its analysis for phase measuring profilometry”. From the thesis I know that Phase measuring profilometry is a method of structured light illumination whose three-dimensional reconstructions are susceptible to error from nonunitary gamma in the associated optical devices. The effects of this distortion diminish with an increasing number of employed phase-shifted patterns, so the more PMP steps, the better the measurement effect. I added this thesis as number 30 to the reference, located at line 175, and added “30. Liu, Kai, et al. "Gamma model and its analysis for phase measuring profilometry." JOSA A 27.3 (2010): 553-562.” in page 19 line 333-334.

7. High speed measurements are expected to give speed values and comparisons.

Response:

Thank you very much for your valuable suggestion. By controlling the speed of the stepper motor to control the real-time movement speed of the cardioid model, the corresponding root-mean-square is calculated according to the same method as the previous real-time comparison. The speed range of the stepper motor is 0~40mm/s. We choose 24mm/s, 30mm/s, and 36mm/s respectively for experimen

Revision:

The height root mean square is made into a table and the results are as follows and added the table and analysis content to page 15 between line 246 and 252:

Table.1 The height root mean square error at different speeds

24mm/s 30mm/s 36mm/s

ZHANG S 0.1435 0.1546 0.1866

Propose 0.0442 0.0513 0.7462

In page 14 line 240-242, we added the sentence of “According to the above comparison method, we select the speed as 24ms/s, 30ms/s, and 36ms/s respectively by controlling the speed of the stepper motor to complete real-time experiment. Then, to calculate the corresponding root mean square value and make a table1 expressed below:”

In page 15 line 244-246, we added the sentence of “It can be observed that the measuring effect becomes worse as the speed increases. The reason for this result is that the higher the speed of the object was, the greater the amplitude of the object’s vibration would be, the more factors that the captured image would interfere with, the more unstable resulting distorted image will be and the worse the measurement effect would be.”

Reviewer #2: The experimental section does not specify if the data is cherry-picked or averaged or repeatable over several frames. Unfortunately, the manuscript contains lots of English errors, and also, references are displayed as "Error! Reference source not found."

1. The experimental section does not specify if the data is cherry-picked or averaged or repeatable over several frames.

Response:

Thank you for raising this question, and sorry for not being able to express it clearly. This article uses video raster projection, and what the camera captures is also a video image. I selected three sets of adjacent two frames of images for three-dimensional measurement. For example, the video images selected in this paper have a total time of 9s, which corresponds to more than 160 frames of images. I selected frames 29-30, 89-90, and 139-140 to complete the three-dimensional measurement.

2. Unfortunately, the manuscript contains lots of English errors.

Response:

Thank you for raising this question, and I am sorry that my English grammatical errors and improper expressions have caused you difficulties in reading. I have corrected the grammar of the thesis from beginning to end.

Revision:

1) In page 2 line 38, “slower speed” has been revised as “a lower speed”.

2) In page 2 line 38, “higher measuring accuracy, wider measurable range” has been revised as “a higher measuring accuracy, a wider range of heights of measuring object”.

3) In page 2 line 41, “faster measurement speed, higher measuring accuracy, and wider measurement application range” has been revised as “a higher measuring speed, higher measuring accuracy, a wider measuring application range”.

4) In page 9 line 172, “Figs 2(c) and 2(d)” has been revised as “Fig 2(c) and Fig 2(d)”.

5) In page 9 line 173, “Figs 2(e) and 2(f)” has been revised as “Fig 2(e) and Fig 2(f)”.

6) In page 9 line 174, “Figs 2(g) and 2(h)” has been revised as “Fig 2(g) and Fig 2(h)”.

7) In page 10 line 181, “Figs 3(c)-(e)” has been revised as “Fig 3(c)–Fig3 (e)”.

8) In page 12 line 208, “Figs 4 (a)-(c)” has been revised as “Fig 4(a)-Fig 4(c)”.

9) In page 12 line 209, “Figs 4(d)-(f)” has been revised as “Fig 4(d)-Fig 4 (f)”.

10) In page 12 line 211, “Figs 4(g)-(i)” has been revised as “Fig 4(g)-Fig 4(i)”.

11) In page 14 line 231, “Figs 5(b) and 5(d)” has been revised as “Fig 5(b) and Fig 5(d)”.

12) In page 9 line 176, page 12 line 216, page 13 line 220 “quasi-true value” has been revised as “quasi-truth”.

13) In page 13 between line 223 and 224, “323th cow” has been revised as “250th column”.

14) In page 14 line 228, “323th cow” has been revised as “250th column”.

15) In page 7 line 131, page 9 line 172 “π/2” has been revised as “ ”.

16) In page 3 line 61, “is” has been revised as “was”.

17) In page 3 line 61, “Firstly” has been revised as “First”.

18) In page 3 line 62, “are” has been revised as “were”.

19) In page 3 line 63, “is” has been revised as “was”.

20) In page 3 line 64, “containing” has been revised as “that covered”.

21) In page 3 line 65, “can” has been revised as “could”.

22) In page 3 line 66, “measured object” has been revised as “object measured”.

23) In page 4 line 73, page 4 line 74, page 4 line 75, page 5 line 104, page 5 line 105, page 5 line 106, page 5 line 107, page 6 line 115, “is” has been revised as “expresses”.

24) In page 6 line 114, “are” has been revised as “denote”.

25) In page 4 line 86, “expressions (2) and (3) and (4)” has been revised as “Eq (2), Eq (3) and Eq (4)”.

26) In page 6 line 114, we add two words “the”.

27) In page 2 line 32-36, The content “FTP requires only one frame of phase shift diagram, retains the fringe frequency in the frequency domain, removes high frequency noise and carrier waves, recovers the fringe pattern from the frequency domain to the spatial domain through inverse Fourier transform, reproduces the fringe field distribution, and finally calculates the phase value of the fringe field [13-14].” has been revised as “FTP requires only one frame of phase shift diagram, It is capable of retaining the fringe frequency in the frequency domain, removing high-frequency noise and carrier waves, recovering the fringe pattern from the frequency domain to the spatial domain via the inverse Fourier transform, reproducing the fringe field distribution, and lastly calculating the phase value of the fringe field [13-14]”.

28) In page 2 line 37-42, The content “Compared with FTP, PMP has slower speed but higher measuring accuracy, wider measurable range, less influence by different reflectivity on the surface of the object, and less influence by drastic changes or fractures on the surface of the object[15-16]. Modern 3D measurement is devoted to the research of faster measurement speed, higher measuring accuracy, and wider measurement application range, so PMP is a better choice[17-21] .” has been revised as “As opposed to FTP, PMP exhibits a lower speed but a higher measuring accuracy, a wider measurable range, less effect by different reflectivity values on object surface, and less effect exerted by drastic variations or fractures on object surface[15-16]. The modern 3D measurement is committed to investigating a higher measuring speed, higher measuring accuracy, as well as a wider measurement application range; thus, PMP is recognized as a more effective choice [17-21].”.

29) In page 3 line 48-49, I corrected the sentence “The technical conditions of the proposed method are limited and the application scope is very narrow[24]”to “The proposed algorithm has the limited technical conditions and a narrow application scope [24]”.

30) In page 3 line 55-57, this sentence is a bit difficult to read, so I corrected “Because the sensitivity of the uniform plane graph to motion blur is much lower than that of the fringe graph, moreover, compared with the traditional 2+1 phase shift algorithm, this method uses less one frame of image and has faster acquisition speed.” to “Since the sensitivity of the uniform plane graph to motion blur is significantly lower than that of the fringe graph, this method, compared with the conventional 2+1 phase shift algorithm, employs less one frame of image and exhibits a higher acquisition speed.”

31) In page 5 line 96-99, The content “If two frames of red and blue uniform background light with the same value as the DC component of the sinusoidal fringe are separately projected to solve the chromaticity imbalance problem caused by the different sensitivity of the CMOS camera to red and blue, then the red color of the same color image in the actual experiment is ignored..” has been revised as “If two frames of red and blue uniform background lights with the identical value as the DC component of the sinusoidal fringe are separately projected to address the chromaticity imbalance problem attributed to the different sensitivities of the CMOS camera to red and blue. Subsequently, the red color of the identical color image in the practical experiment is ignored. To possibly eliminate the chromaticity imbalance attributed the different sensitivities of the CMOS camera to the red and blue lights, two bicolor gratings are designed in advance:”.

32) In page 6 line 124-127, The content “Further analysis shows that and have similar characteristics. It can be seen that the ratio of reflectivity of red channel and blue channel is in constant characteristic. This is caused by the chromaticity imbalance caused by the different sensitivities of CMOS camera to red and blue light.” has been revised as “As indicated from the in-depth analysis, and exhibit the similar characteristics. which demonstrates that the ratio of reflectivity of the red channel and the blue channel exhibits the constant characteristic. This is attributed to the chromaticity imbalance attributed to the different sensitivities of the CMOS camera to the red and blue lights.”.

3. references are displayed as "Error! Reference source not found.

Response:

Thank you so much for your question, and I am sorry that my carelessness caused this format problem. I have corrected the reference number hyperlink problem. 

Revision:

There are three numbering errors in the preview state. 

1) In page 2 line 31, Number “[7-8]” has updated the number hyperlink；

2) In page 2 line 36, Number “[13-14]” has updated the number hyperlink；

3) In page 2 line 42, Number “[17-21]” has updated the number hyperlink；

---

## [Decision Letter · Decision Letter 1]

28 May 2021

PONE-D-21-03909R1

Real-time profilometry by bicolor grating video projection

PLOS ONE

Dear Dr. Cao,

Thank you for submitting your manuscript to PLOS ONE. After careful consideration, we feel that it has merit but does not fully meet PLOS ONE’s publication criteria as it currently stands. Therefore, we invite you to submit a revised version of the manuscript that addresses the points raised during the review process.

We look forward to receiving your revised manuscript.

Kind regards,

Ireneusz Grulkowski, PhD

Academic Editor

PLOS ONE

Reviewers' comments:

Reviewer's Responses to Questions

**Comments to the Author**

1. If the authors have adequately addressed your comments raised in a previous round of review and you feel that this manuscript is now acceptable for publication, you may indicate that here to bypass the “Comments to the Author” section, enter your conflict of interest statement in the “Confidential to Editor” section, and submit your "Accept" recommendation.

Reviewer #2: (No Response)

2. Is the manuscript technically sound, and do the data support the conclusions?

Reviewer #2: No

3. Has the statistical analysis been performed appropriately and rigorously? 

Reviewer #2: No

4. Have the authors made all data underlying the findings in their manuscript fully available?

Reviewer #2: No

5. Is the manuscript presented in an intelligible fashion and written in standard English?

Reviewer #2: No

6. Review Comments to the Author

Reviewer #2: About the data - the authors have indicated in their response to my review that only three pairs of frames were used for the entire paper, out of 9 seconds of video with 160 frames. Some statistical error analysis of the other frames would be desirable, and the raw data should be made available as PLOS ONE's requirement is "PLOS journals require authors to make all data necessary to replicate their study's findings..."

After revision, the manuscript is much more readable. But typos, grammatical mistakes and or omissions remain.

Some suggestions -

1. Line 20 - feasibility and effectiveness of the proposed (method)?

2. Line 32 - add a space after [9-10]

3. Line 33 - It is capitalized after a comma. Either change the comma to a period or don't capitalize.

4. Line 36 - add a period before beginning the sentence with PMP

5. Line 45, Peter L. Wizinowich is in normal case, while in Line 50 and later throughout the document, ZHANG is in ALL CAPS. Normal case would be desirable.

6. Line 51 - "directly projects the uniform plan generated by the computer" - perhaps the authors mean uniform plane illumination generated by the computer? This phrase is also repeated throughout the manuscript.

7. Line 56-57 - employs less one frame of image - perhaps can be phrased as employs one frame less?

8. Line 61-62 - First, only two frames of bicolor patterns substituting for the three frames of essential patterns in the traditional 2+1 3D real-time measuring method... - maybe the authors mean "First, only two frames of bicolor patterns are used, as against the three frames which are essential in the traditional 2+1 3D real-time measuring method. These two frames were encoded and arranged alternately to synthesize a repetitive bicolor grating video.

9. Line 63 - could be better written as "Then the video was projected onto the object and the bicolor..."

10. Line 96 to 98 - The statement starts with an If, but there is no then... perhaps the sentence should have ... separately projected, this can address ... in line 97?

11. Line 123-125 - No capitalization after the period in line 123, and the sentence doesn't make sense. Maybe the authors mean ...Rr(x,y) and Rb(x,y) exhibit similar characteristics. The ratio of reflectivity of the red channel and the blue channel is a constant. This is attributed ....

12. Line 152 - The digital projector used was the PLED-W200 model produced by ViewSonic?

13. Line 153 - The camera used was ...

14. Line 158 - maybe the words "to code in this section" can be omitted?

15. Line 159 - Since we only need to use.....

16. Line 160 - Maybe "At the same time" can be omitted?

17. Line 163 - Maybe have a period after Ic2(x,y) or maybe join the next phrase with an "and" - "... and the chromaticity imbalance coefficient can be..."

18. Line 164 - maybe this line can be omitted?

19. Line 165 - Maybe omit "coded in this section" ?

20. Line 167 - by the CMOS camera

21. Line 175 - replace comma with a period?

22. Line 240-241 - Then, we calculates the corresponding root mean square errors and and made a table as seen in Table 1.

23. In Table 1 - Proposed method instead of Propose

24. Line 247 - It can be observed that the measuring error becomes

25. Line 250 - measurement error

7. PLOS authors have the option to publish the peer review history of their article (what does this mean?). If published, this will include your full peer review and any attached files.

Reviewer #2: No

---

## [Author Response · Author response to Decision Letter 1]

9 Jul 2021

Dear Editor and Reviewers: 

Thank you very much for your excellent editing work of our manuscript entitled “Real-time profilometry by bicolor grating video projection ([PONE-D-21-03909R1]-[EMID:72c44d7d061b8290])”. And we also appreciate the reviewers for their valuable comments about our manuscript. Those comments are all valuable and very helpful for revising and improving our manuscript. We have studied the comments carefully and have made the correction which we hope to be able to meet the approval. The Response to Reviewer and Editor Comments is attached below. Hope our revised manuscript meets the high publication standard set by PLOS ONE.

Sincerely,

Yujiao Zhang, Yiping Cao, Haitao Wu, Haihua An, Xiuzhang Huang Sichuan University

Reviewer #2: 

1. About the data - the authors have indicated in their response to my review that only three pairs of frames were used for the entire paper, out of 9 seconds of video with 160 frames. Some statistical error analysis of the other frames would be desirable, and the raw data should be made available as PLOS ONE's requirement is "PLOS journals require authors to make all data necessary to replicate their study's findings..."

Response:

I am sorry that I only provided three frames of data. In order to be able to provide real-time motion data more accurately, I kept the heart-shaped data and chose a new double-layer rectangular model for experimentation. The multi-frame deformed fringe information and real-time imaging information captured in the experiment are displayed in S1Visualization and S2 Visualizatio, respectively. To further prove the feasibility of real-time exercise, I used this method to measure more complex rotating sectors such as sectors. The multi-frame deformed fringe information and real-time imaging information captured in the experiment are displayed in S3 Visualization and S4 Visualization.

Revision:

1) In page 11 line 201, “heart” has been revised as “double-layer rectangular”.

2) In page 11 line 203, “heart” has been revised as “double-layer rectangular”.

3) In Fig4, “heart” has been revised as “double-layer rectangular”.

4) In page 12 line 209, line 211, line 212, line 213, line 217 “heart” has been revised as “double-layer rectangular”.

5) In page 13 line 226, “heart” has been revised as “double-layer rectangular”.

6) In page 14 line 229, “heart” has been revised as “double-layer rectangular”.

7) In page 12, between line 206 and line 207, I replaced the heart-shaped model in Fig.4 with a double-layer rectangular model

8) In page 13, between line 224 and line 225, I replaced the heart-shaped model in Fig.5 with a double-layer rectangular model.

9) In page 14 line 231-line233, “the results of the two real-time methods are close to the eight-step PMP, however, from Fig 5(b) and Fig 5(d),” has been revised as “the results of the proposed method are close to the eight-step PMP, ZHANG's method is not effective in restoring the middle connection of the double-layer rectangle. From Fig 5(b) and Fig 5(d)”.

10) In page 14 line 241, “0.1456” has been revised as “0.1055”.

11) In page 14 line 241, “0.0153” has been revised as “0.0157”.

12) In page 14, between line 244 and line 246, I replaced the new data in Table1 with a double-layer rectangular model.

Table.1 The height root mean square error at different speeds

24mm/s 30mm/s 36mm/s

ZHANG S 0.0817 0.1055 0.1443

Proposed method 0.0410 0.0517 0.0706

13) In page 11, line 205, I add more deformed patterns with a double-layer rectangular model, and made a S1Visualization of them.

14) In page 12, line 213, I add more reconstructed results with a double-layer rectangular model, and made a S2 Visualization of them.

15) In page 15, line 253, I add more deformed patterns with a more complex sector model, and made a S3 Visualization of them.

16) In page 16, line 261, I add more reconstructed results with a more complex sector model, and made a S4 Visualization of them.

17) In page 15-16, between line 250 and line 263, I add the new reconstruction results with a more complex sector model.

In order to further prove the feasibility of this method, a more complex sector model was measured in real time according to the same method. The bicolor deformed patterns at different times are simultaneously collected by the CMOS camera, and each group of deformed patterns are captured at 30 frames per second (30fps), are shown in Visualization 3. The part of the experimental results is shown in Fig 6.

Fig 6(a)-Fig 6(c) respectively show the bicolor deformed patterns extracted by the real-time online moving sector model under the conditions of state 1, 2, and 3 respectively. Fig 6(d)-Fig 6 (f) respectively show the monochrome deformed patterns extracted by the real-time online moving sector model under the corresponding state conditions. Fig 6(g)-Fig 6(i) respectively show the 3D shapes reconstructed by the real-time online moving sector model under the corresponding time conditions, all reconstructed results are shown in Visualization 4. It can be observed that the 3D reconstruction of the sector model has also been well restored, and its rotating tense can be restored in real time, which further proves the real-time feasibility of the proposed method.

2. Unfortunately, the manuscript contains lots of English errors.

Response:

Thank you for raising this question, and I am sorry that my English grammatical errors and improper expressions have caused you difficulties in reading. I am very grateful for the grammatical errors you pointed out to me, and I corrected the corresponding content according to your suggestions.

Revision:

2) In page 1 line 20, “the proposed” has been revised as “the proposed method”.

18) In page 2 line 32, “profilometry (FTP) [9-10]and” has been revised as “profilometry (FTP) [9-10] and”.

19) In page 2 line 33, “, It” has been revised as “, it”.

20) In page 2 line 36, “[13-14] PMP” has been revised as “[13-14]. PMP”.

21) In page 3 line 51, “uniform plan” has been revised as “uniform plane illumination”.

22) In page 4 line 87, “plane” has been revised as “camera”.

23) In page 3 line 57, “less one frame of image” has been revised as “one frame less.

24) In page 3 line 62-64, “First, only two frames of bicolor patterns substituting for the three frames of essential patterns in the traditional 2+1 3D real-time measuring method were encoded and arranged alternatively to synthesize into a repetitive bicolor grating video” has been revised as “First, only two frames of bicolor patterns are used, as against the three frames which are essential in the traditional 2+1 3D real-time measuring method. These two frames were encoded and arranged alternately to synthesize a repetitive bicolor grating video”.

25) In page 3 line 64-65, “Then, the video was projected onto the object, the bicolor deformed patterns” has been revised as “Then the video was projected onto the object and the bicolor deformed patterns”.

26) In page 5 line 99, “separately projected to address” has been revised as “separately projected, this can address”.

27) In page 6 line 126, “which demonstrates that the ratio of reflectivity of the red channel and the blue channel exhibits the constant characteristic” has been revised as “The ratio of reflectivity of the red channel and the blue channel is a constant”.

28) In page 7 line 153, “VeiwSonic” has been revised as “ViewSonic”.

29) In page 7 line 154, “ImagingSource Company.” has been revised as “The Imaging Source”.

30) In page 7 line 153, “adopts” has been revised as “used was”.

31) In page 7 line 154, “adopts” has been revised as “used was”.

32) In page 8 line 159, to delete “to code in this section” 

33) In page 8 line 160, “it only needs” has been revised as “we only need”.

34) In page 8 line 161, “At the same time, the CMOS” has been revised as “The CMOS”.

35) In page 8 line 163, to add “and” between , and the.

36) In page 8 line 165, to delete “coded in this section”.

37) In page 8 line 165, to delete “Next, the chromaticity imbalance coefficient is used in the bicolor 2+1 3D measurement experiment.”.

38) In page 8 line 167, “through” has been revised as “by the”.

39) In page 9 line 175, “,” has been revised as “.”.

40) In page 14 line 243-244 “Then, to calculate the corresponding root mean square value and make a table1 expressed below:” has been revised as “Then, we calculate the corresponding root mean square errors and made a table as seen in Table 1.”.

41) In Table 1, “Propose” has been revised as “Proposed method”.

42) In page 14 line 246, “effect” has been revised as “error”.

---

## [Decision Letter · Decision Letter 2]

24 Aug 2021

PONE-D-21-03909R2

Real-time profilometry by bicolor grating video projection

PLOS ONE

Dear Dr. Cao,

Thank you for submitting your manuscript to PLOS ONE. After careful consideration, we feel that it has merit but does not fully meet PLOS ONE’s publication criteria as it currently stands. Therefore, we invite you to submit a revised version of the manuscript that addresses the points raised during the review process.

We look forward to receiving your revised manuscript.

Kind regards,

Ireneusz Grulkowski, PhD

Academic Editor

PLOS ONE

Journal Requirements:

Additional Editor Comments (if provided):

Please, correct minor issues as indicated by the reviewer

Reviewers' comments:

Reviewer's Responses to Questions

**Comments to the Author**

1. If the authors have adequately addressed your comments raised in a previous round of review and you feel that this manuscript is now acceptable for publication, you may indicate that here to bypass the “Comments to the Author” section, enter your conflict of interest statement in the “Confidential to Editor” section, and submit your "Accept" recommendation.

Reviewer #3: All comments have been addressed

Reviewer #4: All comments have been addressed

2. Is the manuscript technically sound, and do the data support the conclusions?

Reviewer #3: Yes

Reviewer #4: Yes

3. Has the statistical analysis been performed appropriately and rigorously? 

Reviewer #3: Yes

Reviewer #4: N/A

4. Have the authors made all data underlying the findings in their manuscript fully available?

Reviewer #3: Yes

Reviewer #4: Yes

5. Is the manuscript presented in an intelligible fashion and written in standard English?

Reviewer #3: Yes

Reviewer #4: Yes

6. Review Comments to the Author

Reviewer #3: (No Response)

Reviewer #4: This manuscript proposes a bicolor 2+1 3D real-time measuring method based on bicolor grating video projection. The method is interesting and will be of potential application in manufacturing and robotics. Compared with the 2+1 method proposed by ZHANG S, the proposed method reduces one frame of projected fringes, and only requires to use the red and blue two-channel coded fringes with the smallest overlap range of spectrum response, which reduces the crosstalk of different color channels and improves the 3D measuring speed. As they have addressed the previous major concerns, I think it is suitable to be published in PLOS ONE. Some minor issues should be corrected as follows.

1, The label ‘A’ in Fig 3(c) and the label ‘B’ in Fig 5(a) should be moved to their correct positions.

2, For some sentences, the authors need to pay attention to the usage of tense, such as ‘were’ in line 64 should be ‘are’ and ‘was’ in line 65 should be ‘is’.

3. ‘ ZHANG S' s algorithm’ in both Fig.3 and Fig.5 should be ‘ ZHANG S' algorithm’.

7. PLOS authors have the option to publish the peer review history of their article (what does this mean?). If published, this will include your full peer review and any attached files.

Reviewer #3: No

Reviewer #4: No

---

## [Author Response · Author response to Decision Letter 2]

2 Sep 2021

Dear Editor and Reviewers: 

Thank you very much for your excellent editing work of our manuscript entitled “Real-time profilometry by bicolor grating video projection ([PONE-D-21-03909R2] - [EMID:aa41d53618b9d2b1])”. And we also appreciate the reviewers for their valuable comments about our manuscript. Those comments are all valuable and very helpful for revising and improving our manuscript. We have studied the comments carefully and have made the correction which we hope to be able to meet the approval. The Response to Reviewer and Editor Comments is attached below. Hope our revised manuscript meets the high publication standard set by PLOS ONE.

Sincerely,

Yujiao Zhang, Yiping Cao, Haitao Wu, Haihua An, Xiuzhang Huang Sichuan University

Reviewer 4: 

1. The label ‘A’ in Fig 3(c) and the label ‘B’ in Fig 5(a) should be moved to their correct positions.

Response:

I'm sorry for making a low-level formatting error. I have changed the format of the two pictures as required.

2. For some sentences, the authors need to pay attention to the usage of tense, such as ‘were’ in line 64 should be ‘are’ and ‘was’ in line 65 should be ‘is’.

Response:

Thank you for checking carefully, I have modified as required.

1) In line 63, “were” has been revised as “are”.

2) In line 64, “was” has been revised as “is”.

3. ‘ ZHANG S' s algorithm’ in both Fig.3 and Fig.5 should be ‘ ZHANG S' algorithm’.

Response:

Thank you for your careful inspection. I have checked the keyword in full and modified it.

1) In line 218, “ZHANG S method.” has been revised as “ZHANG S' method”.

2) In line 232, “ZHANG's method” has been revised as “ZHANG S' method”

---

## [Decision Letter · Decision Letter 3]

22 Oct 2021

Real-time profilometry by bicolor grating video projection

PONE-D-21-03909R3

Dear Dr. Cao,

We’re pleased to inform you that your manuscript has been judged scientifically suitable for publication and will be formally accepted for publication once it meets all outstanding technical requirements.

Kind regards,

Ireneusz Grulkowski, PhD

Academic Editor

PLOS ONE

Additional Editor Comments (optional):

Reviewers' comments:

Reviewer's Responses to Questions

**Comments to the Author**

1. If the authors have adequately addressed your comments raised in a previous round of review and you feel that this manuscript is now acceptable for publication, you may indicate that here to bypass the “Comments to the Author” section, enter your conflict of interest statement in the “Confidential to Editor” section, and submit your "Accept" recommendation.

Reviewer #3: All comments have been addressed

2. Is the manuscript technically sound, and do the data support the conclusions?

Reviewer #3: Yes

3. Has the statistical analysis been performed appropriately and rigorously? 

Reviewer #3: Yes

4. Have the authors made all data underlying the findings in their manuscript fully available?

Reviewer #3: Yes

5. Is the manuscript presented in an intelligible fashion and written in standard English?

Reviewer #3: Yes

6. Review Comments to the Author

Reviewer #3: (No Response)

7. PLOS authors have the option to publish the peer review history of their article (what does this mean?). If published, this will include your full peer review and any attached files.

Reviewer #3: No

---

## [Editor Report · Acceptance letter]

15 Nov 2021

PONE-D-21-03909R3 

Real-time profilometry by bicolor grating video projection 

Dear Dr. Cao:

I'm pleased to inform you that your manuscript has been deemed suitable for publication in PLOS ONE. Congratulations! Your manuscript is now with our production department. 

Kind regards, 

on behalf of

Dr. Ireneusz Grulkowski 

Academic Editor

PLOS ONE